# Lengthening adalimumab dosing interval in quiescent Crohn's disease patients: protocol for the pragmatic randomised non-inferiority LADI study

L J T Smits ![ORCID],[1] R W M Pauwels ![ORCID],[2] W Kievit,[3] D J de Jong,[1] A C de Vries,[2] F Hoentjen,[1] C J van der Woude,[2] on behalf of the LADI study group

LJTS and RWMP are joint first authors.
FH and CJvdW are joint senior authors.

¹Department of Gastroenterology and Hepatology, Radboudumc, Nijmegen, The Netherlands
²Department of Gastroenterology and Hepatology, Erasmus Medical Center, Rotterdam, The Netherlands
³Department for Health Evidence, Radboudumc, Nijmegen, The Netherlands

**Correspondence to**
Dr C J van der Woude;
c.vanderwoude@erasmusmc.nl

## ABSTRACT

**Introduction** Adalimumab is effective for maintenance of remission in patients with Crohn's disease (CD) at a dose of 40 mg subcutaneously every 2 weeks. However, adalimumab is associated with (long-term) adverse events and is costly. The aim of this study is to demonstrate non-inferiority and cost-effectiveness of disease activity guided adalimumab interval lengthening compared to standard dosing of every other week (EOW).

**Methods and analysis** The Lengthening Adalimumab Dosing Interval (LADI) study is a pragmatic, multicentre, open label, randomised controlled non-inferiority trial. Non-inferiority is reached if the difference in cumulative incidence of persistent (>8 weeks) flares does not exceed the non-inferiority margin of 15%. 174 CD patients on adalimumab maintenance therapy in long-term (>9 months) clinical and biochemical remission will be included (C-reactive protein (CRP) <10 mg/L, faecal calprotectin (FC) <150 µg/g, Harvey-Bradshaw Index (HBI) <5). Patients will be randomised 2:1 into the intervention (adalimumab interval lengthening) or control group (adalimumab EOW). The intervention group will lengthen the adalimumab administration interval to every 3 weeks, and after 24 weeks to every 4 weeks. Clinical and biochemical disease activity will be monitored every 12 weeks by physician global assessment, HBI, CRP and FC. In case of disease flare, dosing will be increased. A flare is defined as two of three of the following criteria; FC>250 µg/g, CRP≥10 mg/l, HBI≥5. Secondary outcomes include cumulative incidence of transient flares, adverse events, predictors for successful dose reduction and cost-effectiveness.

**Ethics and dissemination** The study is approved by the Medical Ethics Committee Arnhem-Nijmegen, the Netherlands (registration number NL58948.091.16). Results will be published in peer-reviewed journals and presented at international conferences.

**Trial registration numbers** EudraCT registry (2016-003321-42); Clinicaltrials.gov registry (NCT03172377); Dutch trial registry (NTRID6417).

## BACKGROUND

Crohn's disease (CD) is a chronic inflammatory disease of the gastrointestinal tract, characterised by a relapsing and remitting

### Strengths and limitations of this study

► The Lengthening Adalimumab Dosing Interval (LADI) study is the first randomised controlled trial that investigates adalimumab interval lengthening in Crohn's disease patients.
► This pragmatic study is clinically relevant and results can easily be implemented in daily practice.
► The National Crohn and colitis patients organisation is involved and patient-reported outcomes are included.
► The study is not blinded.

disease course. Patients show an abnormal mucosal immune response, resulting from an interplay of genetic susceptibility, environmental factors and the intestinal microflora.[1] Treatment consists of immunosuppressive medication, including monoclonal antibodies that block tumour necrosis factor alpha (anti-TNF); such as infliximab, adalimumab and certolizumab. Adalimumab is a humanised anti-TNF antibody that is effective as both induction and maintenance therapy for CD.[2–4] Adalimumab is administered by subcutaneous (sc) injection and an induction dose of 80 mg (week 0) and 40 mg (week 2) or 160 mg (week 0) and 80 mg (week 2) are generally used, followed by 40 mg every 2 weeks.[5]

Although adalimumab is generally safe, side effects do occur. The risk of (opportunistic) infections is increased, especially in combination with immunosuppressive therapies, most often thiopurines or methotrexate.[6–9] A recently published review on long term safety of adalimumab (n=3606 CD patients) showed a high absolute risk of any infection of 119 events per 100 patient years (PYs) and a risk of serious infection of 6.7/100 PYs in this selected trial-population with relatively low comorbidity.[10] The incidence

rate of injection site reactions (local pain and swelling) was 7.7/100 PYs.[10] In addition, several reports show an increased risk of skin cancer (both melanoma and non-melanoma skin cancer), especially in combination with thiopurines.[6 7 9 11 12] In addition to potential side effects, the costs of adalimumab are significant. Before the introduction of biosimilars, the costs of anti-TNF in the Netherlands were €15 000–30 000 per CD patient annually.[13 14] Anti-TNF including adalimumab is expected to continue to be the main cost driver of CD management for several reasons. First, the number of CD patients is increasing in the Netherlands.[15] Second, recent data stimulate an early use of anti-TNF with an accelerated step-up or top-down approach in combination with treat-to-target (mucosal healing), to prevent bowel damage.[16] Third, the entry of lower cost biosimilars will possibly cause physicians to preferentially prescribe anti-TNF treatment, which will increase its use.[17–19]

Discontinuation of adalimumab therapy in CD patients in stable clinical remission is a clinical strategy that may aid in reducing the risk of side effects, costs and avoid prolonged immunosuppression during a quiescent disease course. However, in a large meta-analysis on individual patient data (n=1330, including the landmark study by Louis *et al*[20]) on cessation of anti-TNF therapy, approximately 38% of the patients had a relapse in 1 year, and 52% after 2 years of follow-up (Pauwels *et al*, unpublished data). Therefore, an alternative strategy of dose reduction of adalimumab rather than discontinuation may be considered. In rheumatoid arthritis (RA), the Dose REduction Strategy of Subcutaneous TNF inhibitors (DRESS) study concluded that disease activity guided dose reduction of anti-TNF is non-inferior and cost-effective, compared with maintaining regular dosing.[21 22] However, extrapolation of these results to CD is questionable, since RA patients generally use different concomitant medication, suffer from different comorbidities and anti-TNF shows different pharmacodynamic characteristics in RA patients.[23 24] In CD, adalimumab dose reduction is uncommon in daily practice. Only two retrospective cohort studies (n=46+40) reported CD patients who used adalimumab 40 mg every 3 weeks (ETW).[25 26] After a median follow-up of 16 and 24 months, respectively 63% and 65% remained in clinical remission.

The aim of this randomised controlled trial is to demonstrate non-inferiority and cost-effectiveness of disease activity guided adalimumab injection interval lengthening compared with standard of care (continued every other week (EOW) dosing) in maintaining remission in CD. In this paper we describe the study design as well as potential pitfalls and outcomes.

## OBJECTIVE
### Primary objective
► To demonstrate non-inferiority of disease activity guided adalimumab injection interval lengthening compared with adalimumab EOW dosing (standard of care) in CD patients in stable disease remission at 48 weeks of follow-up. Non-inferiority is reached if the difference in cumulative incidence of persistent flares not exceeds the non-inferiority margin of 15%. A persistent flare is defined as two of three of the following criteria, persisting for >8 weeks despite dose escalation of adalimumab:
  – Faecal calprotectin (FC) >250 µg/g.
  – C-reactive protein (CRP) ≥10 mg/L.
  – Harvey-Bradshaw Index (HBI) ≥5.

### Secondary objectives
► To report the proportion of patients that had successful interval lengthening, defined as the absence of a disease flare, while treated with adalimumab ETW or every 4 weeks (EFW), at week 48.
► To identify factors that are associated with successful interval lengthening (eg, baseline patient and treatment characteristics, FC, CRP, adalimumab drug levels and antibodies to adalimumab).
► To compare the cumulative incidence of patients with a transient flare (duration ≤8 weeks) between the intervention and control group at week 48.
► To compare the proportion of patients that used budesonide, prednisone or other immunomodulators in order to treat a (transient) flare.
► To compare the proportion of patients in clinical and biochemical remission between the intervention and control group at week 48. Remission is defined as an HBI <5, FC <150 µg/g and CRP <10 mg/L. In case disease activity is assessed with endoscopy or MRI scan, that conclusion overrules our definition.
► To compare inflammatory bowel disease (IBD)-specific quality of life by the Short-IBD Questionnaire (SIBDQ)) between the intervention and control group every 12 weeks during follow-up.
► To compare disease activity by HBI and patient reported outcome (PRO-2) between the intervention and control group every 12 weeks during follow-up.
► To compare medical consumption (by institute for Medical Technology Assessment (iMTA) Medical Consumption Questionnaire (MCQ)) and work productivity (by iMTA Productivity Cost Questionnaire (PCQ)) between the intervention and control group until week 48, in order to calculate the decremental cost-effectiveness ratio of this interval lengthening strategy.
► To compare the rates of (serious) adverse events ((S)AEs) that are (possibly) related to adalimumab and the rates of (S)AEs that are (possibly) related to adalimumab interval lengthening between the intervention and control group, expressed as events/100 PYs of follow-up.
► To compare adalimumab use between the intervention and control group, including the cumulative dose during follow-up, the proportion of patients that uses adalimumab ETW and EFW.

## METHODS

This protocol includes the standard protocol items recommended for interventional trials according to the Standard Protocol Items: Recommendations for Interventional Trials (SPIRIT) guidelines (online supplementary file 1).[27]

### Design

This randomised controlled trial is currently being performed at the departments of Gastroenterology and Hepatology in 21 hospitals in the Netherlands, including both academic and non-academic centres. The aim of the adalimumab interval lengthening strategy is to minimise the amount of adalimumab use while maintaining remission in CD. Therefore, longer adalimumab intervals will be compared with adalimumab EOW in a non-inferiority design (to show the same effect is maintained with a dose reduction strategy), instead of a superiority design, which is used to demonstrate that an intervention leads to superior outcomes than the standard of care. The rationale behind a non-inferiority design is that benefits may be present in other areas (ie, fewer side effects, lower costs) so that the intervention would be preferred if its efficacy is not worse.

The date of the first enrolment was 3 May 2017. The study is approved by the Medical Ethical Committee (METC) Arnhem-Nijmegen (registration number NL58948.091.16). Important protocol modifications are assessed and approved by the METC, and reported to participating investigators. The most recent study protocol version 3.3 (July 2018) is presented in this manuscript. The Lengthening Adalimumab Dosing Interval (LADI) study has been registered at clinicaltrials.gov and the Dutch trial registry. A data safety monitoring board is installed in order to independently assess the efficacy and safety of the study intervention and to monitor the overall conduct of the trial. Prior to enrolment, all patients have to sign informed consent (online supplementary file 2).

### Patient and public involvement

The study was designed in collaboration with the Dutch Crohn's and colitis patient organisation (CCUVN) in order to optimise patient participation. We based our study design on the results of a *biological focus group* by members of the CCUVN. This focus group showed that patients do accept a reduction of the dose of their biological agent. Additionally, based on previous interactions with the CCUVN, we have included patient focused outcomes in our study, such as the quality of life and PRO-2.

### Inclusion and exclusion criteria

All adult CD patients with colonic and/or distal ileal and/or proximal CD, who are treated with adalimumab 40 mg every 2 weeks at a stable dose, at least 9 months in steroid-free clinical remission and not scheduled for CD-related surgery, are eligible for participation.[28] Remission is defined as an HBI <5, FC <150 µg/g and CRP <10 mg/L. The current guidelines from the European Crohn's and Colitis Organisation (ECCO) suggest to use CRP <10 mg/L for the definition of disease remission.[5] Endoscopic assessment prior to enrolment is not mandatory, however if an ileocolonoscopy was performed before the start of the study and demonstrated complete mucosal healing (Simple Endoscopic Score-CD <3 or no ulcerations), an FC <250 µg/g is accepted as inclusion criterium. Permitted concomitant CD therapies are: aminosalicylates, azathioprine, 6-mercatopurine, methotrexate and thioguanine at a stable dose for 12 weeks. Patients with arthralgia will be included, however inflammatory arthritis is an exclusion criterium, as this can provide elevated inflammatory markers. Furthermore, patients with active draining fistulas are excluded. Other exclusion criteria are pregnancy or lactation and other significant medical conditions that might interfere with this study (such as a current/recent malignancy, immunodeficiency syndromes and psychiatric illness), or when it is to be expected that the outcome cannot be measured (short life expectancy, planned major surgery, language issues).

### Study groups

#### Control group

The control group continues maintenance treatment with adalimumab sc 40 mg EOW. Treatment decisions are made at the discretion of the treating physician. Of note, dose reduction beyond 40 mg per 2 weeks is currently not recommended according to national guidelines.[29] Patients follow a standardised protocol based on the tight control/treat-to-target principle in order to maintain low disease activity.[16]

#### Intervention group

Adalimumab interval will be lengthened through a stepwise disease activity guided manner.

Step 1: On inclusion, the interval will be prolonged to ETW.

Step 2: After week 24, patients in remission will lengthen their dosing interval to EFW.

Step 3: If adalimumab interval lengthening leads to a confirmed flare, patients will return to the preceding effective interval (figure 1). If a flare is not objectively confirmed, patients are advised to continue adalimumab in their study-interval. However, interval reduction is accepted if patients really want this as this situation reflects daily clinical practice.

In contrast to the DRESS study, the discontinuation of therapy after successful de-escalation to 40 mg EFW is not implemented in the study protocol.[21] Total follow-up time will be 48 weeks. Follow-up visits and outcome measurements are similar to the control group.

### Cointervention

The use of previously mentioned concomitant medication is allowed and must be documented on the case-report form (CRF) (stating type, dosage and duration). If

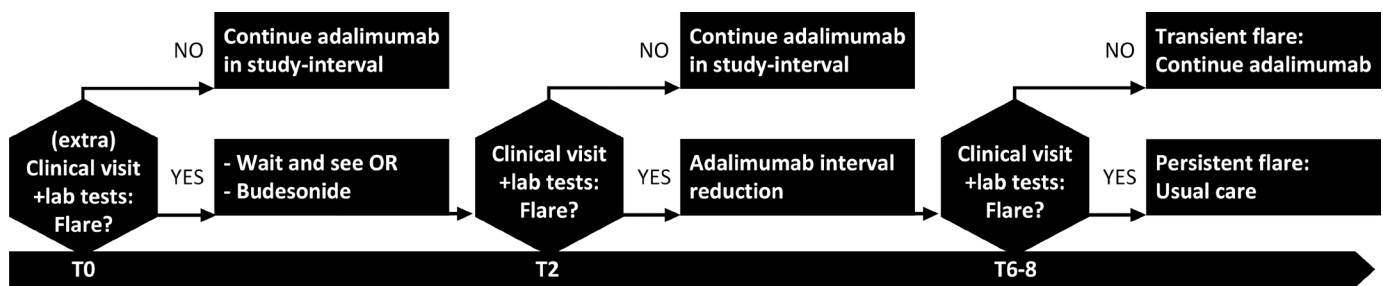

**Figure 1** Protocolised treatment recommendation in case of disease flare. T0: start of possible disease flare, which can occur at any time during follow-up, T2: 2 weeks after T0, T6–8: 6–8 weeks after T0. Lab tests include haemoglobin, leucocytes, thrombocytes, albumin, C-reactive protein, calprotectin.

possible, existing concomitant medication should not be changed during the study.

If patients experience worsening of symptoms in between visits, they must contact the outpatient clinic. For further treatment of the flare, patients in the control arm are referred to their treating physician. In the intervention arm, patients will return to the preceding effective adalimumab dosing interval (figure 1). The decision to start concomitant therapy remains at the discretion of the treating physician.

### Secondary outcome measurements
#### Quality of life
For assessment of quality of life, we will use the SIBDQ, which is a validated and disease-specific questionnaire.[30]

#### Patient reported disease activity
We will use the only validated IBD patient-reported outcome measure, 'PRO-2', consisting of reported diarrhoea and abdominal pain.[31]

#### Factors associated with successful dose reduction
Factors which are possibly related to successful dose reduction include: baseline patient and treatment characteristics, adalimumab drug levels (µg/mL) and antibodies (AU/mL), clinical (physician global assessment, HBI) and laboratory results (FC (µg/g), CRP (mg/L), haemoglobin (g/L), leucocytes ($10^9$/L), platelets ($10^9$/L), albumin (g/L)).

#### Safety
AEs and SAEs are registered during follow-up. All SAEs are reported to the METC Arnhem-Nijmegen.

#### Cost-effectiveness
The impact of dose reduction on the quality of life of patients will be assessed by the EuroQuol-5D (EQ-5D) at 24 and 48 weeks following randomisation, compared with baseline. The EQ-5D utility will be used to derive a quality-adjusted life year (QALY) estimate for each patient according to the trapezium rule.[32 33]

### Assessments
Enrolled patients will visit the outpatient clinic every 24 weeks. If preferred by the patient or treating physician, the evaluations at week 12 and 36 can take place

as outpatient clinic visit as well. Every 12 weeks, laboratory tests (eg, FC, CRP, haemoglobin and albumin) will be performed. At week 0, 24 and 48 serum samples are stored for measurement of adalimumab drug levels and antibodies to adalimumab. Additionally, patients in both arms will be interviewed via telephone every 6 weeks in between clinical visits to assess for adverse events, symptoms and potential disease activity. If such an interview suggests a disease flare, patients must visit the outpatient clinic in order to undergo complete disease activity assessment and laboratory and FC tests. If patients have a flare at week 48, disease activity will be monitored until disease remission, in order to define the flare as persistent- or transient flare. In addition, study questionnaires are automatically sent via Castor every 12 weeks. During follow-up, patients register the adalimumab injection dates in a study-diary and bring this to the outpatient clinic every visit to evaluate adherence to adalimumab. An overview of all visits and assessments is depicted in table 1 and figures 1 and 2.

### Randomisation, allocation concealment, stratification
Patients are randomised by the research physician using a computer-generated randomisation system (Castor). Castor uses a validated variable block randomisation model with block sizes of 6, 9 and 12. Patients will be randomised in a 2:1 ratio for the intervention or the control group, respectively. We chose 2:1 randomisation to stimulate patient inclusion, as patients have a higher chance to randomise for the intervention group. Furthermore, more determinants can be included in a prediction model for successful dose reduction if the intervention group is larger. Patients will be stratified on comedication use (yes/no), as the incidence of flares could possibly be different with or without comedication use. Comedication includes azathioprine, 6-mercaptopurine, 6-thioguanine, methotrexate. Both patients and physicians are un blinded, as we aim to represent daily practice during this pragmatic study.

### Sample size
The null hypothesis in non-inferiority studies is that the intervention is inferior compared with the control arm by more than the non-inferiority margin. The alternative hypothesis is that the intervention is not worse than

**Table 1** SPIRIT schedule of enrolment, interventions and assessments

| | Study period | | | | | | | | | | | | |
|---|---|---|---|---|---|---|---|---|---|---|---|---|---|
| | Enrolment | Allocation | Follow-up | | | | | | | | | | Extra |
| Timepoint | $-t_1$ | 0 | $w_0$ | $w_6$ | $w_{12}$ | $w_{18}$ | $w_{24}$ | $w_{30}$ | $w_{36}$ | $w_{42}$ | $w_{48}$ | | $w_e$ |
| **Enrolment** | | | | | | | | | | | | | |
| Eligibility screen | X | | | | | | | | | | | | |
| Informed consent | X | | | | | | | | | | | | |
| Allocation | | X | | | | | | | | | | | |
| **Interventions** | | | | | | | | | | | | | |
| *Intervention: lengthening adalimumab dosing interval* | | | ←————————————————————→ | | | | | | | | | | |
| *Control: adalimumab every other week* | | | ←————————————————————→ | | | | | | | | | | |
| **Assessments** | | | | | | | | | | | | | |
| *Medical history* | X | X | | | | | | | | | | | |
| *Laboratory tests\** | | | X | | X | | X | | X | | X | | X |
| *Faecal calprotectin* | | | X | | X | | X | | X | | X | | X |
| *Storage of serum samples* | | | X | | | | X | | | | X | | X |
| *Concomitant medication* | | | X | X | X | X | X | X | X | X | X | | X |
| *(Serious) adverse events* | | | X | X | X | X | X | X | X | X | X | | X |
| *Physician global assessment* | | | X | X | X | X | X | X | X | X | X | | X |
| *HBI* | | | X | | X | | X | | X | | X | | X |
| PRO-2, *IBD-Q and EQ5D* | | | X | | X | | X | | X | | X | | |
| *iMTA MCQ, PCQ* | | | X | | X | | X | | X | | X | | |

\*Haemoglobin, leucocytes, thrombocytes, albumin, C-reactive protein.
EQ-5D, EuroQuol-5D; HBI, Harvey Bradshaw Index; IBD-Q, Inflammatory Bowel Disease Questionnaire; iMTA MCQ, institute for Medical Technology Assessment Medical Consumption Questionnaire; PCQ, Productivity Cost Questionnaire; PRO-2, patient reported outcome-2; SPIRIT, Standard Protocol Items: Recommendations for Interventional Trials.

the control by more than the non-inferiority margin. Therefore, if the null hypothesis is rejected, the alternative hypothesis that the intervention is non-inferior is accepted.[34] Based on an extrapolation of data from the DRESS study and results from a real-life CD cohort in Leuven, an estimated 15% of patients will experience the primary outcome (persistent flare) in the control arm. In the Leuven cohort, 41/156 (26%) patients discontinued adalimumab due to loss of response, despite adalimumab dose escalation.[21 35] The latter

26% was adjusted to an expected 15% for our cohort because the follow-up time in our cohort concerns 12 rather than 20 months, and our cohort is a preselected cohort of patients in long and stable remission rather than a cross-sectional cohort. In non-inferiority analyses, one-sided testing is used. Applying one sided testing, an alpha of 0.05 ($Z\alpha$=1.64), power 1-beta 0.8 ($Z\beta$=0.84), a non-inferiority margin of 15% and randomisation ratio of 2:1 intervention versus control resulted in n=105 and n=53 for intervention and control arm, respectively.

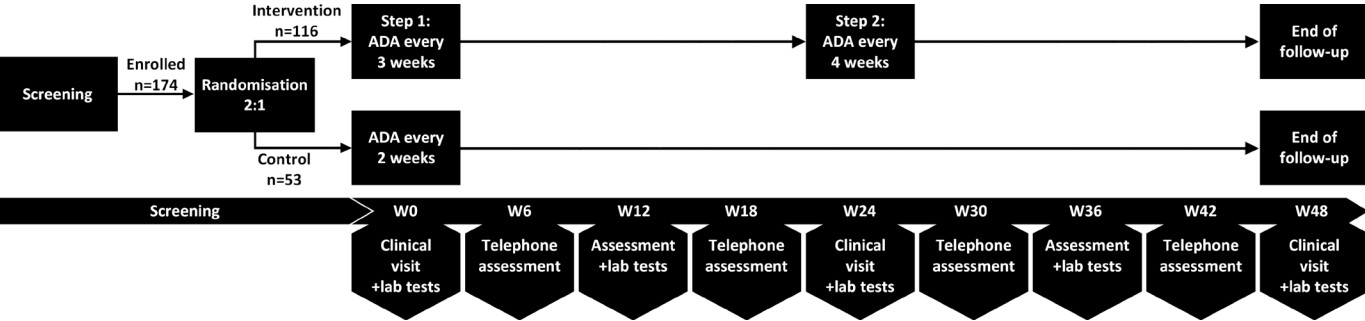

**Figure 2** Schematic presentation of the trial design. ADA, adalimumab; W0, week 0; W6, week 6 and so on. Lab tests include haemoglobin, leucocytes, thrombocytes, albumin, C-reactive protein, calprotectin.

Accounting for a 10% drop-out, 174 patients have to be included in total.

A non-inferiority margin of 15% means a maximum difference in persistent flare of 15% between the usual care and intervention group. We believe this strikes an acceptable balance between the potential harms of flare, and the benefits of dose reduction (fewer injections, potential for reduced risk of side effects and cost-savings). The large Nor-Switch trial also used a non-inferiority margin of 15% for disease worsening during follow-up.[36] Based on this example, discussions in our study-group and approval of the protocol by the Dutch Organisation for Health Research and Development, we believe this margin is appropriate. The DRESS study used a non-inferiority margin of 20%. Although side effects/SAEs of adalimumab seem comparable in RA versus IBD, rheumatologists probably accept a higher proportion of flares because there are more alternative biological therapies available, thus a loss of effect of one biological therapy might be given less weight in RA.[6]

### Planned data analysis

The primary outcome; cumulative incidence of persistent flares will be expressed as proportions in both groups. A CI for the difference between study groups will be determined (adjusted for comedication use at baseline using the Cochran-Mantel-Haenszel procedure, as this variable is used for stratification in the randomisation process).[37] The upper limit of the CI will be compared with the non-inferiority margin. We will use both intention to treat and per protocol analyses, as the latter is considered the most conservative analysis for non-inferiority trials.[38] Patients in the interval lengthening group are included in the per protocol analyses if they: lengthened the adalimumab interval at least to 3 weeks, regardless whether they returned to a preceding effective interval in case of a disease flare. Patients in the control group are included in the per protocol analyses if they: used adalimumab EOW without consistent interval lengthening, incidental postponement of an injection during infection or around holidays is allowed. Descriptive patient (and treatment) baseline variables will be summarised as means±SD, medians with IQRs or percentages, depending on the type of measurement. Gender, body mass index, age, prior medication for CD, disease duration, Montreal classification, IBD-related surgical history, comorbidity, inflammatory parameters including HBI, FC, CRP, adalimumab drug levels and antibodies to adalimumab will be reported.

The secondary continuous outcomes HBI, SIBDQ, PRO-2, adalimumab drug levels and antibody levels at 48 weeks will be analysed by either Student's t-test or Mann-Whitney U test depending on the type of distribution of the data. In addition, the course over time for several continuous outcomes measured at multiple time points (every 12 weeks) will be analysed using repeated measures analyses in which the outcome can be corrected for the baseline value of the specific outcome and potential confounding factors. The number of (S)AEs that are (possibly) related to adalimumab or to adalimumab interval lengthening will be reported as rates, defined as events/100 PYs of follow-up; details of these (S)AEs will be provided. In the intervention group, patient characteristics and clinical features will be analysed to predict a persistent flare. A prediction model will be developed and fitted using a univariable selection based on a p value <0.2 and a multivariable approach with backward selection. Predictive accuracy will be determined by the area under the receiver operating curve. A two-sided p value of <0.05 is considered statistically significant. All statistical analyses will be performed by using IBM SPS Statistics V.25.0.

### Cost-effectiveness analysis

The cost analysis consists of two main parts. First, at patient level, volumes of care related to the CD care and anti-TNF therapy will be measured by means of the iMTA MCQ. This questionnaire measures all relevant healthcare related costs like outpatient visits at any medical specialist, hospitalisations and imaging procedures. Loss of productivity due to illness or recovery in patients below the age of 65 years will be estimated based on patient reported absences from paid (or unpaid) labour measured with the PCQ. The second part of the cost analysis consists of determining the cost prices for each volume of consumption. The standard cost prices from the 'Dutch Guidelines for Cost Analyses' and www.medicijnkosten.nl will be used. For units of care where no standard prices are available real costs prices will be determined on the basis of full cost pricing. Productivity losses will be valued by means of the friction cost method. In the end volumes of care will be multiplied with the cost prices for each volume of care to calculate costs. Because we anticipate non-inferiority of the dose reduction strategy, we will primarily analyse cost-savings: direct medical cost as well as total costs (medical and non-medical costs) will be compared between intervention and control group. A possible small but acceptable loss of effect can be incorporated in the analyses by determining a decremental cost-effectiveness ratio (DCER) by dividing the difference in costs by the difference in QALYs between the groups. The DCER expresses with how much money a loss of one QALY is compensated. If this amount is high the decision makers may be willing to accept a loss of effect. Uncertainty in the DCER will be non-parametrically determined using bootstrap techniques (1000 replications). Results from this analysis will be presented in a scatter plot and willingness to pay (or accept) curve. Furthermore, the Net Monetary Benefit per patient will be calculated for different levels of willingness to accept (WTA) in euro's per QALY, using the formula: WTA×effect (difference in QALY)−costs. This results in the net amount of money saved, when the possible loss of QALY is corrected for, using different WTA levels per QALY.

### Ethics and dissemination

The study is approved by the Medical Ethics Committee Arnhem-Nijmegen, the Netherlands (registration number NL58948.091.16). Data of all participating centres will be

collected by electronic CRFs and monitored following good clinical practice (GCP) guidelines. The collected data will be entered in Castor, an electronic database set up for clinical trials (https://www.castoredc.com). Data will be coded and kept based on the rules for GCP by certified personnel. Results will be published in peer-reviewed journals and presented at international conferences.

## DISCUSSION

Dose reduction of adalimumab in CD patients with stable disease may provide similar disease control but reduction of adverse events and costs. With this pragmatic, non-inferiority study design we aim to evaluate the outcomes of this strategy. Only two small retrospective studies reported on adalimumab 40 mg ETW in CD patients, and demonstrated that approximately two thirds of patients could safely de-escalate adalimumab.[25 26] However, no prospective randomised data are available to confirm these data. Prior studies have investigated the effect of discontinuation of anti-TNF therapy in CD.[20 39–41] Previous clinical trials on withdrawal of anti-TNF after a period of prolonged remission in CD patients showed a relatively consistent profile of 42% relapses after anti-TNF cessation within 1 year of follow-up.[39–41] Louis *et al* identified risk factors for disease flare after discontinuation of infliximab in CD patients who used infliximab and thiopurine combination therapy for at least 1 year. Risk factors for relapse included male sex, high leucocyte counts, high CRP, high FC and low levels of haemoglobin.[20] The multicentre randomised CEASE trial (Diagnostic tool to safely CEASE anti-TNF therapy in Crohn's disease, ZonMw project number 848101009) will further investigate cessation of anti-TNF. As cessation of anti-TNF therapy is a different research question with different outcome measures, uncertainty remains on factors that are associated with successful adalimumab interval lengthening and the LADI study will provide useful information for daily clinical practice.

We decided to assess non-inferiority with regard to persistent flares (persisting >8 weeks independent of treatment changes such as adalimumab dose escalation) since these are the most relevant clinical outcomes in this setting. Temporary flares (persisting <8 weeks) that resolve after appropriate treatment are less difficult to manage and are likely to occur as an acceptable result of searching for the optimal individualised treatment interval. Temporary flares will still function as relevant secondary outcome in our trial. For the definition of a flare, two consecutive measurements demonstrating two out of three of the following criteria; FC >250 µg/g, CRP ≥10 mg/L, HBI ≥5 are required. As it has been shown that flares are frequently temporary and occur and sometimes disappear without regimen change, a flare is only considered a flare if it is confirmed two times. For this composite endpoint we preferred to incorporate the HBI instead of the Crohn's Disease Activity Index on account of accessible clinical implementation in daily practice.

In addition, FC and CRP are non-invasive, cheap and widely available biomarkers of disease activity.[42] Furthermore, FC correlates to endoscopic disease activity.[43 44] Recently, it was shown that an increase in FC can precede on the onset of clinical symptoms.[45] Indeed, due to our definition of a flare, patients without clinical symptoms can also fulfil the definition of a (biochemical) flare. In addition, the requirement of an elevation in inflammatory markers at two time points allows for the exclusion of confounders such as *Clostridium difficile* infection and use of 'non-steroidal anti-inflammatory drugs' (NSAIDs).

We decided not to include endoscopy outcomes in the inclusion criteria or primary endpoint. An endoscopic procedure is a burden for patients due to the invasive procedure and the intensive preparation. In addition, we aimed for study results that may be easily implemented in daily practice. Instead of an endoscopy, we used a combination of surrogate markers of inflammation including HBI, CRP and FC to determine clinical remission. A protocolised treatment is advised when a flare occurs (figure 1). However, treatment choices are not mandatory and bridging therapy (including steroids) is left to the discretion of the treating physician.

For the study design, a blinded design was considered, but the development, costs and administration of placebo injections would create a formidable barrier for the study. Furthermore, an un-blinded (pragmatic) design fits best with the current ideas about the external validity of cost-effectiveness studies. This design mirrors the real-life setting which is also not blinded, with respect to costs and effects. In general, an unblinded study design could result in information and attribution bias, for example, flares in patients in whom the dose is reduced would possible be reported sooner. Because this will not lead to an underestimation of the drawbacks of a dose reduction strategy, this bias was accepted.

Our trial will provide important insights in addition to the risk of recurrence as well as the risk of persistent flares. For example, we will collect valuable series of drug measurements of adalimumab. Although the DRESS study did not show predictive value of drug levels for the success of dose reduction, daily IBD practice does apply measurement of drug levels. It is possible that drug levels at baseline, either low or high, may predict successful dose reduction.

From a societal perspective, it is important to improve the cost-effectiveness of IBD healthcare. Patients with chronic inflammatory diseases use expensive medication for many years and there is a growing amount of new (expensive) drugs that will soon be implemented in daily clinical care. In RA and psoriasis, dose reduction trials in adalimumab treated patients are performed and in RA the feasibility of this strategy was already demonstrated and results from a Dutch nation-wide psoriasis trial will follow soon.[21 46] The recent introduction of biosimilars of adalimumab will further aid in cost reduction but the new costs of this therapy will still remain significant.

Therefore, cost savings due to dose reduction will remain relevant.

In conclusion, we designed a pragmatic randomised controlled trial to assess the non-inferiority of a strategy of adalimumab dose reduction in CD patients. Accurate prediction of successful tapering may aid in reduction of adverse events and costs to further improve care for CD patients.

**Acknowledgements** This study is supported by the Dutch Initiative on Crohn and Colitis (ICC), a nationwide network of IBD centres that aims at initiating and facilitating IBD research. We thank Dr A A den Broeder, rheumatologist at St. Maartenskliniek Nijmegen, for his invaluable help in designing the study.

**Collaborators** Centres and lead investigators that collaborate as "LADI study group": Albert Schweitzer Hospital, Dordrecht, The Netherlands; F.H.J. Wolfhagen, MD PhD, Department of Gastroenterology. Amphia Hospital, Breda, The Netherlands; A.G.L. Bodelier, MD PhD, Department of Gastroenterology. Amsterdam University Medical Centre-location AMC, Amsterdam, The Netherlands; M. Löwenberg, MD PhD, Department of Gastroenterology. Amsterdam University Medical Centre-location VUmc, Amsterdam, The Netherlands; N. de Boer, MD PhD, Department of Gastroenterology. Bernhoven Hospital, Uden, The Netherlands; I.A.M. Gisbertz, MD PhD, Department of Gastroenterology. Canisius-Wilhelmina Hospital, Nijmegen, The Netherlands; A.C.I.T.L. Tan, MD PhD, Department of Gastroenterology. Erasmus Medical Centre, Rotterdam, The Netherlands; C.J. van der Woude, MD PhD, Department of Gastroenterology. Elisabeth-Tweesteden Hospital, Tilburg, The Netherlands; M.W.M.D. Lutgens, MD PhD, Department of Gastroenterology. Flevoziekenhuis Hospital, Almere, The Netherlands; R.C. Mallant-Hent, MD PhD, Department of Gastroenterology. Franciscus Gasthuis & Vlietland, Rotterdam, The Netherlands; R.L. West, MD PhD, Department of Gastroenterology. Ikazia Hospital, Rotterdam, The Netherlands; P.C.J. ter Borg, MD PhD, Department of Gastroenterology. Jeroen Bosch Hospital, Den Bosch, The Netherlands; T.E.H. Römkens, MD PhD, Department of Gastroenterology. Leiden University Medical Centre, Leiden, The Netherlands; A.E. van der Meulen, MD PhD, Department of Gastroenterology. Maastricht University Medical Centre+, Maastricht, The Netherlands; M. Pierik, MD PhD, Department of Gastroenterology. Maxima Medical Centre, Veldhoven/Eindhoven, The Netherlands; P.J. Boekema, MD PhD, M.L. Verhulst, MD PhD, Department of Gastroenterology. Medisch Spectrum Twente, Enschede, The Netherlands; M.G.V.M. Russel, MD PhD, Department of Gastroenterology. Onze Lieve Vrouwe Gasthuis, Amsterdam, The Netherlands; J. Jansen, MD PhD, Department of Gastroenterology. Radboud University Medical Centre, Nijmegen, The Netherlands; F. Hoentjen, MD PhD, Department of Gastroenterology. Reinier de Graaf Hospital, Delft, The Netherlands; S.V. Jansen, MD, Department of Gastroenterology. University Medical Centre Utrecht, Utrecht, The Netherlands; B. Oldenburg, MD PhD, Department of Gastroenterology. Zuyderland Hospital, Heerlen/Sittard, The Netherlands; M.J.L. Romberg-Camps, MD PhD, A.A. van Bodegraven, MD PhD, Department of Gastroenterology. Data safety monitoring board: R.J.M Brüggemann, PharmD PhD, Department of Pharmacy, Radboud University Medical Centre, Nijmegen, The Netherlands (chair). J. in t Hout, PhD, Department for Health Evidence, Radboud University Medical Centre, Nijmegen, The Netherlands. R.J. de Knegt, MD PhD, Department of Gastroenterology and Hepatology, Erasmus Medical Centre, Rotterdam, The Netherlands. Independent expert: E.T.T.L. Tjwa, MD PhD, Department of Gastroenterology and Hepatology, Radboud University Medical Centre, Nijmegen, The Netherlands.

**Contributors** FH and CJvdW designed the LADI study. WK provided statistical expertise in clinical trial design. ACdV, DJdJ, RWMP, LJTS critically reviewed the study design. Study coordinators LJTS and RWMP ensure daily study management. LJTS and RWMP drafted the manuscript and all authors read, revised and approved the final manuscript. Principal investigators: FH, MD PhD and CJvdW, MD PhD. LJTS and RWMP share first authorship. FH and CJvdW share last authorship.

**Funding** The investigator initiated LADI study is supported by the Netherlands Organisation for Health Research and Development (ZonMw, Healthcare Efficiency programme, grant number 848015002). ZonMw is part of the Netherlands Organisation for Scientific Research (NWO). Sponsor: Radboud University Medical Centre P.O. Box 9101, 6500 HB Nijmegen, The Netherlands.

**Competing interests** DJdJ received consulting fees from Synthon Pharma, Abbvie and MSD, and travel fees from Falk Pharma, Takeda, Abbvie, MSD, Ferring, Vifor Pharma and Cablon Medical. ACDV has participated in advisory board and/or received financial compensation from the following companies: Jansen, Takeda, Abbvie and Tramedico. FH has served on advisory boards or as speaker for Abbvie, Janssen-Cilag, MSD, Takeda, Celltrion, Teva, Sandoz and Dr Falk, and received unrestricted funding from Dr Falk, Janssen-Cilag, Abbvie and Celgene. CJvdW received grant support from Falk Benelux and Pfizer; received speaker fees from AbbVie, Takeda, Ferring, Dr Falk Pharma, Hospira, Pfizer; and served as a consultant for AbbVie, MSD, Takeda, Celgene, Mundipharma and Janssen.

**Patient and public involvement** Patients and/or the public were involved in the design, or conduct, or reporting, or dissemination plans of this research. Refer to the Methods section for further details.

**Patient consent for publication** Not required.

**Provenance and peer review** Not commissioned; externally peer reviewed.

**Data availability statement** The dataset generated during the LADI study is available on reasonable request.

**Author note** Coordinating centers are: Radboud University Medical Centre, Department of Gastroenterology and Hepatology, Nijmegen, The Netherlands. Erasmus Medical Centre, Department of Gastroenterology and Hepatology, Rotterdam, The Netherlands.

**ORCID iDs**
L J T Smits http://orcid.org/0000-0002-4090-0738
R W M Pauwels http://orcid.org/0000-0002-2118-5687

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
