## [Reviewer comments · BMJ Open]

ARTICLE DETAILS

TITLE (PROVISIONAL)	Lengthening Adalimumab Dosing Interval in quiescent Crohn's disease patients: protocol for the pragmatic randomised non-inferiority LADI study.
AUTHORS	Smits, Lisa; Pauwels, Renske Wilhelmina Maria; Kievit, Wietske; Jong, D; De Vries, Annemarie; hoentjen, F; van der Woude, Janneke

VERSION 1 – REVIEW

REVIEWER	Takayuki Matsumoto Iwate Medial University, Japan
REVIEW RETURNED	08-Dec-2019

GENERAL COMMENTS	1. Full term of LADI should be indicated in the abstract.2. Abstract, line at around 24. CRP\leq10mg/ml should be CRP\leq10mgmg/L.3. Page 7, line 8. (standard of case) should be deleted.4. Page 7, line at around 25. EFW should not be abbreviated here.5. Page 7, line at around 44. Full term for MRI should not be capitalaized.6. Page 8, last line. "every four weeks" should be deleted.7. Please use one of "enrollment" and "enrolment".8. Page 12, line at around 9. Do patients really visit the clinical every 24 weeks?9. Procedures for monitoring adherence to adalimumab should be indicated.10. Page 17, bottom line. The sentence should "such as Clostridium difficile infection use of NSAIDs".
--

REVIEWER	TADAKAZU HISAMATSU Department of Gastroenterology and Hepatology, Kyorin University School of Medicine
REVIEW RETURNED	19-Jan-2020

GENERAL COMMENTS	I am honored to have the opportunity to review this trial protocol. 1. As the author states, concerns about the side effects in maintenance therapy by biologics and increasing medical costs are major challenges in IBD management. Some patients who have been able to maintain remission with currently prescribed dosing regimens could be able to extend the dosing interval. Ideal is a tailor-made treatment based on drug concentration measurements, but it is not realistic. Therefore, this reviewer believes the significance of this study, which attempted to extend the administration interval of biologics.
--

	2. Trial registration: EudraCT: 2016-003321-42. Registered on 26 September 2016. Clinicaltrials.gov: NCT03172377. Registered on 1 June 2017 The study has already been registered and started as described above. Since this is a review of a protocol paper, this reviewer wants to make sure that data collection has not been completed. 3. The study is designed as an open-label, non-inferiority trial compared with a standard dosing regimen as a control. This reviewer considers that it is reasonable to conduct the study in an open-label manner in daily clinical practice from the viewpoint of feasibility of the study. This reviewer also agrees with the authors that they did not require an endoscopic assessment as enrollment criteria or outcomes. 4. Concomitant medications are being stratified and the protocol is well designed. One concern is that if patients with small intestinal Crohn's disease are enrolled in one group, the incidence of surgery due to stenosis may be biased. Similar concerns exist for silent perianal fistula.
--	---

VERSION 1 – AUTHOR RESPONSE

Reviewer: 1

Reviewer Name

Takayuki Matsumoto

Institution and Country

Iwate Medical University, Japan

Please state any competing interests or state 'None declared':

No CIO to disclose.

Please leave your comments for the authors below

1. Full term of LADI should be indicated in the abstract.

Thank you, we added the full term (Lengthening Adalimumab Dosing Interval) in the abstract.

2. Abstract, line at around 24. CRP≤10mg/ml should be CRP≤10mgmg/L.

Thank you, we changed it into CRP≤10mg/L and checked the remainder of the manuscript.

3. Page 7, line 8. (standard of case) should be deleted.

We agree, this text is deleted.

4. Page 7, line at around 25. EFW should not be abbreviated here.

Thank you, we used the full term only the first time this abbreviation is introduced (page 6)

5. Page 7, line at around 44. Full term for MRI should not be capitalized.

Thank you, the capitals are removed.

6. Page 8, last line. "every four weeks" should be deleted.

Thank you (see 4.)

7. Please use one of "enrollment" and "enrolment".

Thank you, we chose to use "enrollment" and checked the entire manuscript.

8. Page 12, line at around 9. Do patients really visit the clinical every 24 weeks?

Yes, these visits are combined with regular visits and patients are used to visit their gastroenterologist at the clinic at least once every 6 months.

9. Procedures for monitoring adherence to adalimumab should be indicated.

We added this in the protocol, section Assessments page 11:

All patients register the adalimumab injection dates in a study-diary and bring this to the outpatient clinic every visit to evaluate adherence to adalimumab.

10. Page 17, bottom line. The sentence should "such as Clostridium difficile infection use of NSAIDs".

Thank you, we changed the sentence.

Reviewer: 2

Reviewer Name

TADAKAZU HISAMATSU

Institution and Country

Department of Gastroenterology and Hepatology, Kyorin University School of Medicine

Please state any competing interests or state 'None declared':

None declared.

Please leave your comments for the authors below

I am honored to have the opportunity to review this trial protocol.

1. As the author states, concerns about the side effects in maintenance therapy by biologics and increasing medical costs are major challenges in IBD management. Some patients who have been able to maintain remission with currently prescribed dosing regimens could be able to extend the dosing interval. Ideal is a tailor-made treatment based on drug concentration measurements, but it is not realistic. Therefore, this reviewer believes the significance of this study, which attempted to extend the administration interval of biologics.

Thank you for emphasizing the relevance of our trial design.

2. Trial registration: EudraCT: 2016-003321-42. Registered on 26 September 2016.

Clinicaltrials.gov: NCT03172377. Registered on 1 June 2017

The study has already been registered and started as described above. Since this is a review of a protocol paper, this reviewer wants to make sure that data collection has not been completed.

Indeed, data collection is not completed.

3. The study is designed as an open-label, non-inferiority trial compared with a standard dosing regimen as a control. This reviewer considers that it is reasonable to conduct the study in an open-label manner in daily clinical practice from the viewpoint of feasibility of the study. This reviewer also

agrees with the authors that they did not require an endoscopic assessment as enrollment criteria or outcomes.

Thank you very much, we indeed constructed this study to closely follow clinical practice.

4. Concomitant medications are being stratified and the protocol is well designed. One concern is that if patients with small intestinal Crohn's disease are enrolled in one group, the incidence of surgery due to stenosis may be biased. Similar concerns exist for silent perianal fistula.

Thank you for this remark. We chose to also include patients with small intestinal Crohn's disease and perianal fistulas in this study, in order to increase the external validity of the results from this pragmatic clinical study. We assume that these cases will be equally divided over the two study groups by randomization. We considered adding more stratification factors, however every factor has a negative effect on the natural randomization process. Therefore we chose not to add small intestinal Crohn's disease or perianal fistulas as stratification-factors.

VERSION 2 – REVIEW

REVIEWER	TADAKAZU HISAMATSU Department of Gastroenterology and Hepatology, Kyorin University School of Medicine
REVIEW RETURNED	09-Feb-2020

GENERAL COMMENTS	The manuscript is improved.
-----------------------------